# Retrieval of 500 m Aerosol Optical Depths from MODIS Measurements over Urban Surfaces under Heavy Aerosol Loading Conditions in Winter

**Shikuan Jin [1], Yingying Ma [1,2,*], Ming Zhang [3], Wei Gong [1,2], Oleg Dubovik [4], Boming Liu [1], Yifan Shi [1] and Changlan Yang [1]**

1   State Key Laboratory of Information Engineering in Surveying, Mapping and Remote Sensing, Wuhan University, Wuhan 430079, China; jinsk@whu.edu.cn (S.J.); weigong@whu.edu.cn (W.G.); liuboming@whu.edu.cn (B.L.); yifanshi@whu.edu.cn (Y.S.); yangchl@whu.edu.cn (C.Y.)
2   Collaborative Innovation Center for Geospatial Technology, Wuhan 430079, China
3   Hubei Key Laboratory of Critical Zone Evolution, School of Geography and Information Engineering, China University of Geosciences, Wuhan 430074, China; zhangm@cug.edu.cn
4   CNRS, UMR 8518-LOA-Laboratoire d'Optique Atmosphérique, Univ. Lille, F-59000 13 Lille, France; oleg.dubovik@univ-lille.fr
*   Correspondence: yym863@whu.edu.cn

**Abstract:** Moderate Resolution Imaging Spectroradiometer (MODIS) aerosol products are used worldwide for their reliable accuracy. However, the aerosol optical depth (AOD) usually retrieved by the operational dark target (DT) algorithm of MODIS has been missing for most of the urban regions in Central China. This was due to a high surface reflectance and heavy aerosol loading, especially in winter, when a high cloud cover fraction and the frequent occurrence of haze events reduce the number of effective satellite observations. The retrieval of the AOD from limited satellite data is much needed and important for further aerosol investigations. In this paper, we propose an improved AOD retrieval method for 500 m MODIS data, which is based on an extended surface reflectance estimation scheme and dynamic aerosol models derived from ground-based sun-photometric observations. This improved method was applied to retrieve AOD during heavy aerosol loading and effectively complements the scarcity of AOD in correspondence with urban surface of a higher spatial resolution. The validation results showed that the retrieved AOD was consistent with MODIS DT AOD (R = ~0.87; RMSE = ~0.11) and ground measurements (R = ~0.89; RMSE = ~0.15) from both the Terra and the Aqua satellite. The method can be easily applied to different urban environments affected by air pollution and contributes to the research on aerosol.

**Keywords:** aerosol optical depth; urban surfaces; remote sensing; heavy aerosol loading

## 1. Introduction

Aerosols consist of particles with different chemical compositions, optical properties, shape, and size, which have significant impacts on the global climate changes [1]. They influence the earth's radiation budget balance, not only by directly scattering incoming solar light, but also indirectly by participating in cloud formation as cloud condensation nuclei, which can also affect precipitation substantially [2–5]. In addition, increasing emissions of anthropogenic aerosol can cause air pollution and pose a threat to public health [6,7]. The accurate acquisition of aerosol information can contribute to the estimation of aerosol particle concentration, investigation of climate changes, monitoring of aerosol environmental pollution, and other studies.

Satellite remote sensing technology has the advantages of a large area coverage, a long effective time, and a low relative cost, and it has become one of the main methods used to obtain aerosol distribution information at a global scale [8]. A number of aerosol retrieval algorithms have been proposed and improved based on different satellite multi-channel sensors, such as the Moderate Resolution Imaging Spectroradiometer (MODIS) [9–11], the Advanced Very High Resolution Radiometer [12], the Ozone Monitoring Instrument [13], and the Advanced Himawari Imager [14,15]. The MODIS sensors onboard the Terra and Aqua satellites of the Earth Observation System have a swath width of ~2000 km, covering the entire globe nearly daily. The aerosol optical depth (AOD) products released by MODIS have been widely verified, recognized, and used. The MODIS operational land Dark Target (DT) aerosol retrieval algorithm is based on two main assumptions [16]: (1) the existence of a linear relationship between the short-wave infrared reflectance of the surface (2.12 μm) and the visible bands corresponding to the dense dark vegetation pixels; (2) the mixing of several predefined aerosol types. The DT algorithm performs stably when applied to the observation of land with a low apparent reflectance under clear-sky conditions. By comparing with aerosol robotic network sun-photometer data, more than 60% of AOD derived from the DT algorithm fell within expected error (EE) bounds at a global scale [17].

China, the largest developing country in the world, is currently suffering from severe air pollution due to its rapid industrialization and urbanization [18]. Increase in anthropogenic aerosol emissions causes air pollution and changes the composition of aerosols, affecting the inversion of AOD by passive satellites. Several researchers have been focusing on heavily polluted areas of North China. Wang et al. [19] suggested that about a third of the satellite data obtained from the Visible Infrared Imaging Radiometer Suite had been removed incorrectly due to heavy aerosol loading conditions. Li et al. [20] indicated the default aerosol models included in the DT algorithm may not be suitable and proposed a haze AOD retrieval method for several haze events based on haze detection and a global 3-D atmospheric chemical transport mode. Yan et al. [21] developed an enhanced haze aerosol retrieval algorithm that can work not only on hazy days, but also on normal weather days. However, most of these research projects have been interested in large areas characterized by low spatial resolution: there is a lack of studies on AOD retrieval from satellite measurements with a relatively high resolution over urban surfaces.

In recent years, Central China has also faced severe air pollution due to its rapid economic development and massive emissions of anthropogenic aerosol particles. Wuhan is a megacity located on the Jianghan Plain of Central China, which has 11 million permanent residents and covers more than 8000 km$^2$, with an urbanization area of ~80%. A large amount of water vapor is released by rivers and lakes (Figure 1) located around the city, resulting in high humidity all year around. These conditions lead to the hygroscopic growth of the aerosol particles, influencing the gas-particle conversion rates, and the formation of local haze [22–24]. These adverse factors, coupled with the great uncertainty in the estimation of relatively highly reflectant urban surfaces, negatively affect the retrieval of aerosol from satellite measurements. The successful retrieval of AOD under such conditions is challenging. Notably, long-term cloud cover and frequent haze events reduce the number of effective satellite observations during winter [25]; hence, it is important to retrieve as much aerosol information as possible from the limited data available. An efficient aerosol retrieval under heavy loading conditions is imperative and it can be useful for investigations on regional climate change, air pollution control, and aerosol.

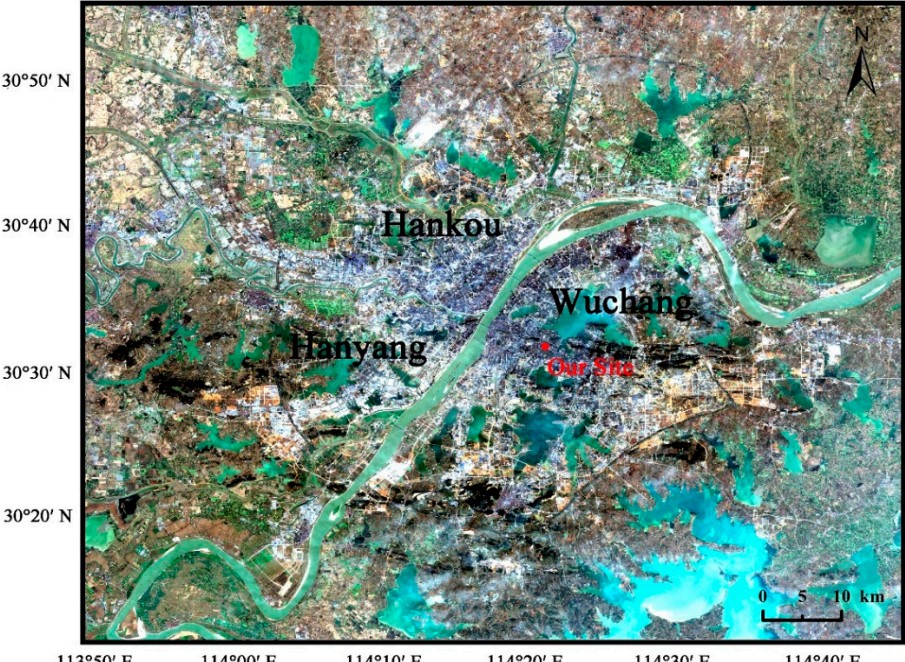

**Figure 1.** True color satellite image shows the real urban landform derived from Landsat-8 OLI measurements (https://www.usgs.gov) on 17 December 2017. The Yangtze River system divides Wuhan into three major towns: Hankou, Hanyang, and Wuchang. The red dot represents our observation sites installed on Wuhan University.

In this study, we used improved aerosol models and a modified DT algorithm to retrieve AOD from 500 m MODIS measurements conducted in winter, taking Wuhan as an example. A relatively high spatial resolution can help to reveal changes in the aerosols present in urban environments and provide useful information for future studies [26]. In addition, FY-3D, China's self-developed satellite is equipped with the Medium Resolution Spectral Imager II, which has been operating for one year [27]. This sensor was designed to be similar to that of MODIS and will be used as an important tool for climate change and atmospheric pollution investigations in the next few years. The present study also lays the foundation for the retrieval of aerosol information from FY-3D satellite images.

## 2. Data, Measurements, and Methods

### 2.1. Site and Satellite Data

The aerosol observation station was installed on the roof of the State Key Laboratory of Information Engineering in Surveying, Mapping, and Remote Sensing (114°21′E, 30°32′N) at Wuhan University, where the main source of aerosol is urban activities. Aerosol optical properties used in the study are acquired from measurements of a CIMEL sun–sky radiometer (CE-318) following the methods proposed by Dubovik [28,29]. This instrument is calibrated annually using the China Meteorological Administration Aerosol Remote Sensing Network to ensure the reliability of the data [30].

The daytime Terra/Aqua MODIS images and products are acquired from NASA's Level-1 and Atmosphere Archive and Distribution System (LAADS) Distributed Active Archive Center (DAAC) (https://ladsweb.modaps.eosdis.nasa.gov/search/).

### 2.2. Aerosol Model

Aerosols are difficult to characterize uniformly at a global scale due to their large spatiotemporal variability and relatively short lifecycle in the atmosphere [31]. They are variable in the Wuhan area, influenced by differences of meteorological factors, local anthropogenic emissions, and long-distance dust transport. The variations of aerosol in Wuhan can be divided into three main stages according

to previous research [24,25,32]: (1) On clean days, the coarse mode particles, mainly from local raising mineral dust, are predominant. (2) With the development of air pollution, both absorbing and non-absorbing fine mode particles increase, and they become dominant. (3) During extreme air pollution periods, the number and size of non-absorbing fine mode particles continue to increase. It is probably due to the increase of the rate of gas-particle conversion and particle hygroscopic growth under high humidity conditions. Based on the above understanding, we classified aerosols' optical properties, using measurements of the CE-318 from 2012 to 2016, employing a K-means clustering approach to obtain two AOD dependence aerosol models [33]: a relative scattering aerosol model (SAM) and an absorbing aerosol model (AAM), as shown in Figure 2. The two aerosol models were relatively independent and the input characteristic parameters in cluster analysis were single scattering albedo (SSA) at 670 nm and asymmetry factor at 440 nm to avoid creating a dust cluster from ultraviolet absorption (at near 440 nm) [33,34].

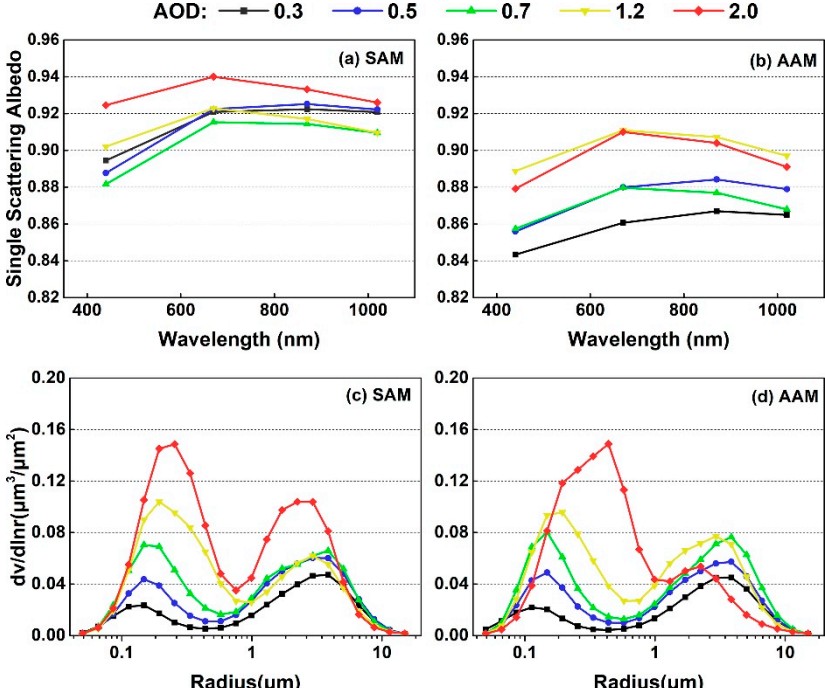

**Figure 2.** Single scattering albedo (**a**,**b**) and aerosol volume size distribution (**c**,**d**) of scattering aerosol model (SAM) and absorbing aerosol model (AAM) were calculated from CE-318 measurements by K-means clustering approach.

Table 1 displays the basic parameters (at AOD = 0.5) of the SAM, AAM, and MODIS DT algorithm aerosol models [33] and the classical continental aerosol model [35]. The AAM shows high absorption compared with Developing and Urban-Ind models in the MODIS DT algorithm which are generally used in China. The dynamic aerosol models describe the changes of aerosol more accurately and are beneficial in retrieving AOD from passive satellite images, especially in heavy aerosol loading conditions in which the aerosol signal occupies a larger proportion of the total signal. Although the classical continental aerosol model is suggested not to be conducive to retrieval aerosols because of its independence from AOD, it is still useful to describe natural continental background aerosols. In addition, the increase in aerosol particle size due to hygroscopic growth is underestimated in MODIS DT algorithm aerosol models under high humidity conditions such as in Central and Southern China. Those inappropriate assumptions of aerosols may result in inaccurate and failed AOD retrievals.

**Table 1.** Comparing SAM and AAM with MODIS DT algorithm aerosol models and the continental model at an aerosol optical depth (AOD) of 0.5.

| | Model | $r_f$ | $r_c$ | $v_f$ | $v_c$ | $\sigma_f$ | $\sigma_c$ | Refractive Index |
|---|---|---|---|---|---|---|---|---|
| This study | SAM | 0.159 | 2.188 | 0.072 | 0.114 | 0.494 | 0.631 | 1.495–0.007i |
| | AAM | 0.144 | 2.210 | 0.074 | 0.113 | 0.482 | 0.644 | 1.501–0.012i |
| MODIS DT algorithm | Developing | 0.155 | 3.269 | 0.096 | 0.092 | 0.442 | 0.778 | 1.430–0.007i |
| | Smoke | 0.139 | 3.922 | 0.094 | 0.065 | 0.423 | 0.763 | 1.510–0.020i |
| | Urban-Ind | 0.162 | 3.396 | 0.097 | 0.060 | 0.440 | 0.841 | 1.420–0.006i |
| | Dust | 0.147 | 2.200 | 0.043 | 0.326 | 0.682 | 0.574 | 1.502–0.002i |
| Continental | WATE | 0.180 | | 3.050 | | 1.090 | | 1.530–0.006i |
| | DUST | | 17.600 | | 7.360 | | 1.090 | 1.530–0.008i |
| | SOOT | 0.050 | | 0.105 | | 0.690 | | 1.750–0.430i |

*Note.* r, v, and σ represent aerosol volume modal radius, total volume, and standard deviation, respectively. Subscript f and c mean fine-mode and coarse-mode aerosols. The refractive index shows the absorbing and scattering characteristics of aerosol particles at 670 nm.

### 2.3. Surface Reflectance

The estimation of surface reflectance from satellite images has been a difficult task mainly due to large uncertainties from inhomogeneous land cover and surface properties. Kaufman et al. [36] indicated firstly that surface reflectance of MODIS band 1 (0.66 µm) and band 3 (0.47 µm) were highly linearly correlated with band 7 (2.12 µm) over dense vegetation lands, due to absorption of solar light by chlorophyll and liquid water. As the band 7 is generally considered insensitive to fine particle aerosols, the surface reflectance of two visible bands can be estimated, whereas the linear relationship is found to be variable at different satellite observation angles and underlying surfaces. Therefore, Levy et al. [16] updated the linear relationship by adding a scattering angle and shortwave infrared Normalized Difference Vegetation Index (NDVI$_{SWIR}$) calculated by band 5 (1.24 µm) and band 7. The update was widely considered useful and was operated in MODIS aerosol products over land (C005-L). Over urban surfaces, however, the linear relationship was found to change again due to different surface properties compared with dense vegetation lands. Gupta et al. [37] re-calculated the linear relationship and considered it as a function of satellite scattering angle, NDVI$_{SWIR}$, and urban cover percentage (UP) in order to reduce the estimation uncertainty of surface reflectance under complex urban conditions.

$$\text{UP} = \sum_1^n S_i / S_{total} \cdot LT_i \tag{1}$$

The calculation of UP is shown in Equation (1) by which we extend the surface reflectance estimation scheme from 10 km to 500 m resolution MODIS images by inverse distance weighting approach. The $LT_i$ means land type at the $i$th pixel obtained from the MODIS Land Cover Type product (MCD12Q1) which is a yearly product at 500 m resolution and provides five different classification schemes based on measurements from both MODIS sensors [38]. Once the land type is considered an urban surface, the value of $LT_i$ is set to 1 (otherwise 0). $n$ is the number of land cover pixels falling into the range of apparent reflectance pixels controlled by longitude and latitude. $S_i$ and $S_{total}$ represent inverse distance of apparent reflectance pixel with the $i$th land type pixel and sum of all inverse distance, respectively. Namely, $S_i/S_{total}$ is the weighting for $i$th land cover pixel. Then, the urban surface reflectance can be estimated according to the scheme proposed by Gupta et al. [37] after determination of the UP.

### 2.4. Retrieval Strategy

The normalized apparent reflectance $\rho_{TOA}$ received by satellite at a particular wavelength can be approximated to Equation (2) [39]:

$$\rho_{TOA}(\mu_s, \mu_v, \varnothing) = \rho_0(\mu_s, \mu_v, \varnothing) + \frac{T(\mu_s)T(\mu_v)\rho_s(\mu_s, \mu_v, \varnothing)}{[1 - \rho_s(\mu_s, \mu_v, \varnothing)S]} \tag{2}$$

where, $\rho_0(\mu_s, \mu_v, \varnothing)$ is the equivalent reflectance of atmosphere path, $T(\mu_s)T(\mu_v)$ is the upward and downward atmospheric transmissivity, $\rho_s(\mu_s, \mu_v, \varnothing)$ is the surface binomial reflectance, and $S$ is the hemispheric reflection of the atmospheric lower boundary. $\mu_s$, $\mu_v$, and $\varnothing$ are the satellite observation geometry parameters solar zenith angle, satellite zenith angle, and relative azimuth angles, respectively. Each term on the right side can also be considered as a function of the aerosol, except for the surface reflectance.

In order to calculate the theoretical apparent reflectance, a vector second simulation of a satellite signal in the solar spectrum (6S) radiative transfer code [40–42] is employed. The latest version 2.1 was available and released in November 2018 (http://6s.ltdri.org/). It contains several classic aerosol models (such as continental, ocean, urban, and dust) and allows user-defined aerosol particles with different compositions or optical properties. Moreover, compared to previous scalar versions, the vector 6S radiative transfer code presets more wavelength nodes and vertical layers of atmosphere and also accounts for polarization which contributes to avoiding substantial retrieval errors in blue wavelength [43].

The first step in the AOD retrieval is screening of suitable pixels to remove the influence of cloud, snow, water, and shadow. The cloud mask was acquired from MODIS cloud mask products (MOD/MYD35) which defined the "probably clear" (95% cloud free) and "confident clear" (99% cloud free) dataset [44]. The Normalized Difference Snow Index (NDSI) calculated by MODIS band 4 (0.55 μm) and band 6 (1.64 μm) was used to detect snow cover. If the NDSI is greater than 0.4, it typically indicates the presence of snow [45]. The urban water (rivers or lakes) changed little in a year and we removed it based on land cover type from MCD12Q1 mentioned above. Shadows are usually due to clouds, terrain undulation, and high-rise buildings and have low reflectance in near infrared wavelengths [46]. Therefore, we used a simple threshold test (band 7 < 0.03) to delete shadow pixels.

The second step is running the vector 6S radiative transfer code to obtain the atmospheric parameters shown in Equation (2). Input parameters included customized aerosol models (SAM and AAM), the continental aerosol model, observational geometry conditions, altitude of target and sensor, and spectral information of MODIS. Last, we obtained six LUTs from the three aerosol models and two MODIS spectral channels (band 1 and band 3).

The last step was retrieving the AOD from the MODIS data. Prior to this, a gas absorption correction was performed on the initial apparent reflectance image, following the method of Patadia et al. [47]. The precipitable water vapor data was acquired from MODIS water vapor product (MOD/MYD05) and the total ozone column was obtained from the Ozone Monitoring Instrument. Due to the study area being located on a low elevation plain (~30m), the variation of Rayleigh optical thickness was negligible. Then the AOD was retrieved by comparing observed and simulated apparent reflectance as shown in Equations (3) and (4):

$$\left(\rho_{Blue} - \rho^*_{Blue}\right)/\rho_{Blue} + \left(\rho_{Red} - \rho^*_{Red}\right)/\rho_{Red} \le \varepsilon \tag{3}$$

$$\rho^* = \eta \cdot \rho^{LUT}_{cust} + (1 - \eta) \cdot \rho^{LUT}_{cont} \tag{4}$$

where $\rho$ is the apparent reflectance observed by MODIS, $\rho^*$ is the corresponding theoretical apparent reflectance, and $\varepsilon$ is the retrieval error. Subscript *Red* and *Blue* means MODIS band1 and band 3. $\rho^*$ is calculated by mixing $\rho^{LUT}_{cust}$ and $\rho^{LUT}_{cont}$ which are the theoretical apparent reflectance calculated by LUTs, respectively, using customized (SAM or AAM depends on $\varepsilon$) and continental aerosol models. $\eta$ represents the external linear mixing ratio. Finally, the value of AOD is determined by the minimum $\varepsilon$.

## 3. Results and Discussions

### 3.1. Retrieval Results

In order to assess our AOD retrieval method which is proposed here, both MODIS sensors images were processed over Wuhan from the last three winters. The annual average AOD in winter over

Wuhan is shown in Figure 3. High values of AOD (~1.0) were concentrated in suburban areas rather than city centers in the winters, mainly including the north, southwest, and southeast of the city. This phenomenon was most likely related to the distribution of local emission sources as heavy industry moved to the periphery of the city for alleviating the air pollution. People living in suburbs also tended to have more burning behavior (biomass and fossil fuels instead of electricity) during winters, which increased the emissions of anthropogenic aerosol particles. Compared with 2016, the value of AOD in 2017 and 2018 showed a slight decline overall, suggesting the air quality improved. It is worth noting that the AOD increased observably in 2018 over the central city (Figure 3c, the black dotted box) which was probably attributed to increased human activity such as subway construction and urban renewal for the forthcoming Military World Games 2019. Retrieval of AOD with a high resolution shows more aerosol information over the city, and it lays the foundation for further particle pollution investigation.

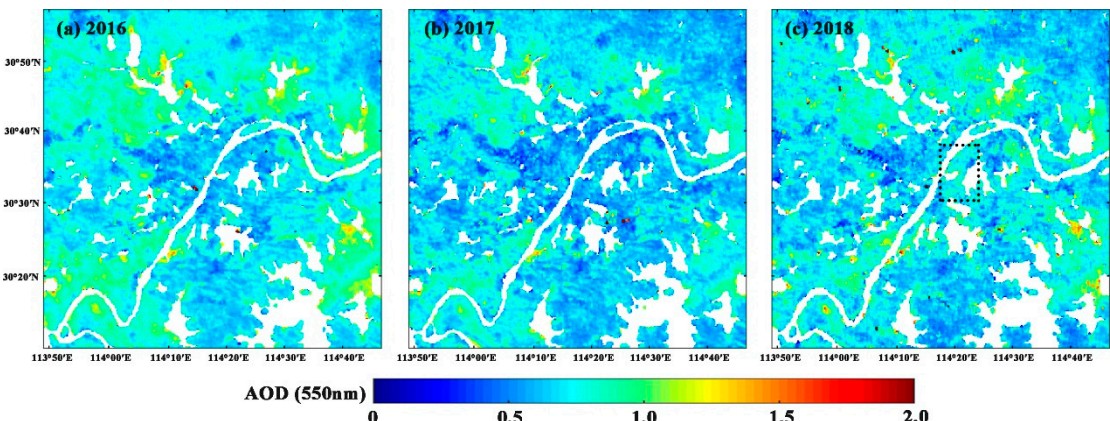

**Figure 3.** Spatial distribution of average AOD in winter from 2016 to 2018 over Wuhan. The black dotted box in 2018 (**c**) shows the increase in AOD over the central city.

### 3.2. Validation

The uncertainties of satellite aerosol retrieval are mainly owed to surface reflectance estimation, aerosol model, and suitable pixels selection [48]. The sun–sky radiometer can observe atmosphere directly and is the most accurate way to measure aerosol, including aerosol extinction and their optical properties. Figure 4 shows the validations of the results by comparison with in-situ measurements. The retrieval results are discussed from five evaluation indexes: fitting equation, correlation coefficient (R), root-mean-square error (RMSE), EE ($0.05 + 0.15\tau$), and match number of retrieval (N). Overall, our results have the highest EE (~70%) and the lowest RMSE (~0.15) in both MODIS sensors compared with DT and "Deep Blue" (DB) products. The R also stays at a relatively high level, just below the DT 3 km resolution products. It suggests that our AOD retrieval method achieves better accuracy in the study area. Although more than 70% of the points fall in the EE bounds, the low slope (~0.7) reveals that the retrieval results are generally overestimated in the case of high AOD. A relatively low retrieval efficiency of both the 3 km and 10 km DT products is found in the region, showing a whole overestimation of AOD (almost out of EE) and the lower N which is almost less than half of the DB products and our results. It is probably due to inappropriate assumptions in aerosol models of DT algorithm which is designed for a global scale. The absorption of aerosols is underestimated in winters, resulting in inaccurate and failed AOD retrievals in the heavy aerosol loading conditions. In addition, compared with the DT algorithm, our method has effectively increased the N in AOD retrieval which is needed to investigate aerosol over city. N is highest in DB products, whereas the RMSE is large (up to 0.516 in Terra MODIS) and the fitting slope is low (~0.6), indicating that the retrieval results are not reliable enough.

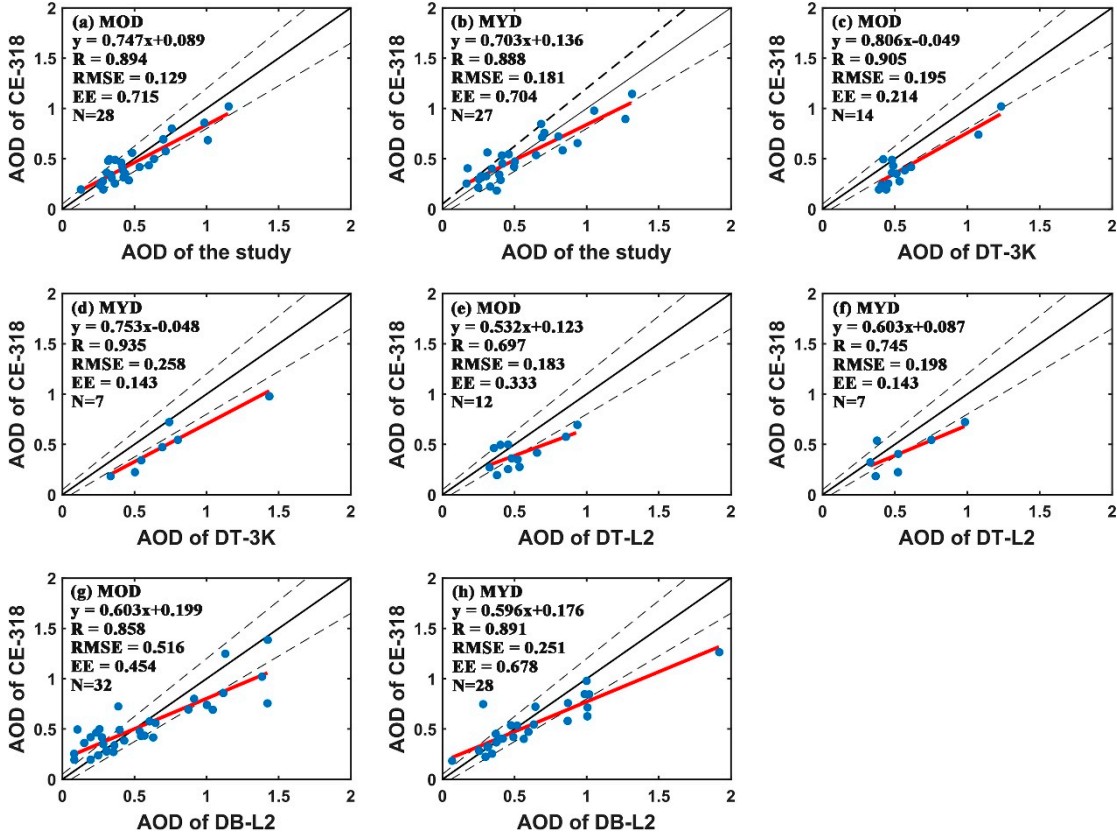

**Figure 4.** Scatter plot of AOD at 550 nm versus that of CE-318 measurements at our site. The MOD and MYD mean Terra (**a**,**c**,**e**,**g**) and Aqua (**b**,**d**,**f**,**h**) satellites, respectively. The DT-3K, DT-L2, and DB-L2 are the value of AOD at 3 km "Dark Target", 10 km "Dark Target", and 10 km "Deep Blue" acquired from MODIS aerosol products (MOD/MYD04). The red full line is a linear fitting line, and the black dotted line is the expected error line.

To further evaluate our results of AOD retrieval on different spatial and temporal scales, we calculated the mean value at the same spatial resolution and compared it with three MODIS products, as shown in Figure 5. R (~0.9) and RMSE (~0.1) show a great consistency of our results with 3 km DT products. The slopes of fitting are less than 1, revealing that the overestimation of DT algorithm was partially relieved. The same phenomenon was also found in 10 km DT products, whereas a relatively poor correlation (~0.7 R) was found with DB products. The DB products have a trend to underestimate AOD under clear conditions and overestimate AOD in heavy loading aerosol periods, as the low slope (~0.6) shows. It is also consistent with the comparison between DB products and CE-318 measurements in Figure 4. In summary, the method in this study not only improves the accuracy but also increases the number of effective retrieval pixels. It is important for the follow-up investigation of particulate pollution in the city.

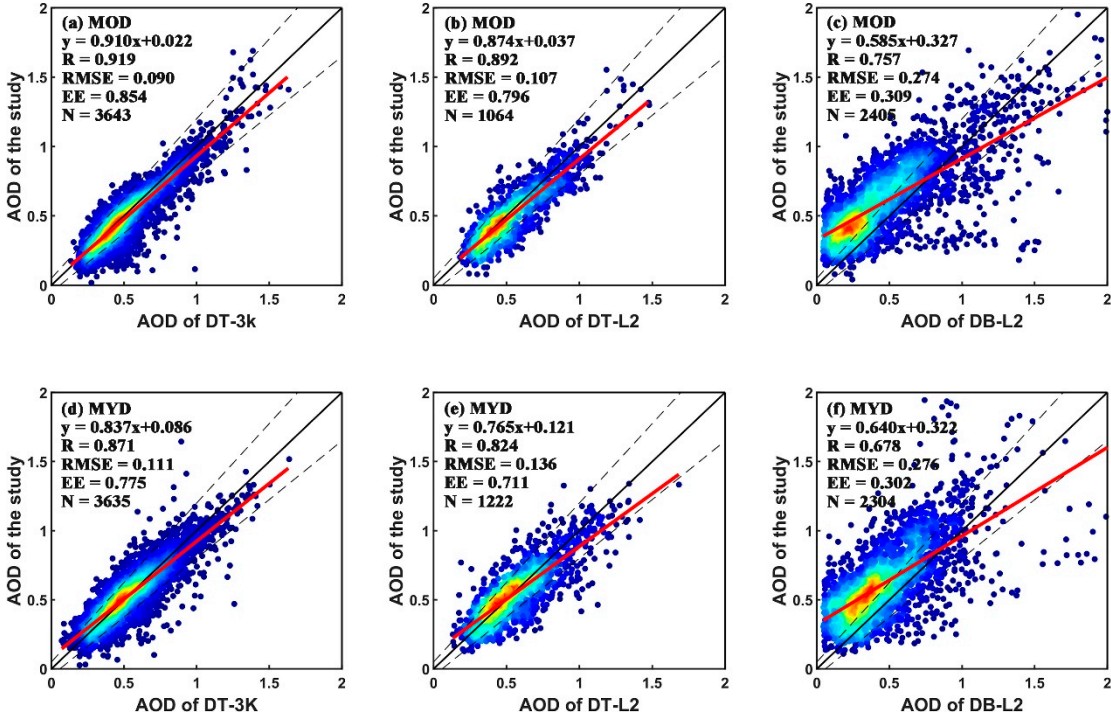

**Figure 5.** Comparing our results (ordinate) with three MODIS products. The MOD and MYD mean sensors on Terra (**a**–**c**) and Aqua (**d**–**f**) respectively. The DT-3K, DT-L2, and DB-L2 are the 3 km "Dark Target", 10 km "Dark Target", and 10 km "Deep Blue". The red full line is a linear fitting line, and the black dotted line is the expected error line.

*3.3. Case Analysis*

Several cases are also selected to better evaluate our results of AOD retrieval from Terra and Aqua MODIS sensor measurements under different conditions, as shown in Figure 6. It is noted that the true color images are composed of three MODIS bands (1,3, and 4) and each band underwent a separate linear normalization enhancement for a clearer presentation. Normally, to our eyes, the clouds are bright white, the vegetation and soil are dark, and the urban surfaces appear off-white. The city spreads out on the eastern and western sides of the river including a large densely populated area where the AOD value of the MODIS operational DT algorithm is missing. Our AOD retrieval results efficiently complement the missing values within the city limits and are very consistent with DT 3 km products in terms of spatial distribution over land. Under the heavy loading aerosol conditions of ~1.0 AOD (such as Figure 6b,d), more details of the aerosol spatial distribution are displayed especially in the region with extremely high values of AOD (red dots) which we focus on. The partial high values of AOD in the suburban areas due to straw burning, setting off fireworks, and winter heating, which are generally forbidden in the city center to prevent the worsening of air pollution, may reveal sources of aerosol emission. This information leads us to further study regional aerosols and control air pollution in the further plan. In addition, our retrieval method can also produce reasonable results during relatively low AOD periods (Figure 6a,c). Sometimes the aerosol is concentrated in the north of Wuhan, suggesting the influence of aerosol from Northern China (Figure 6a).

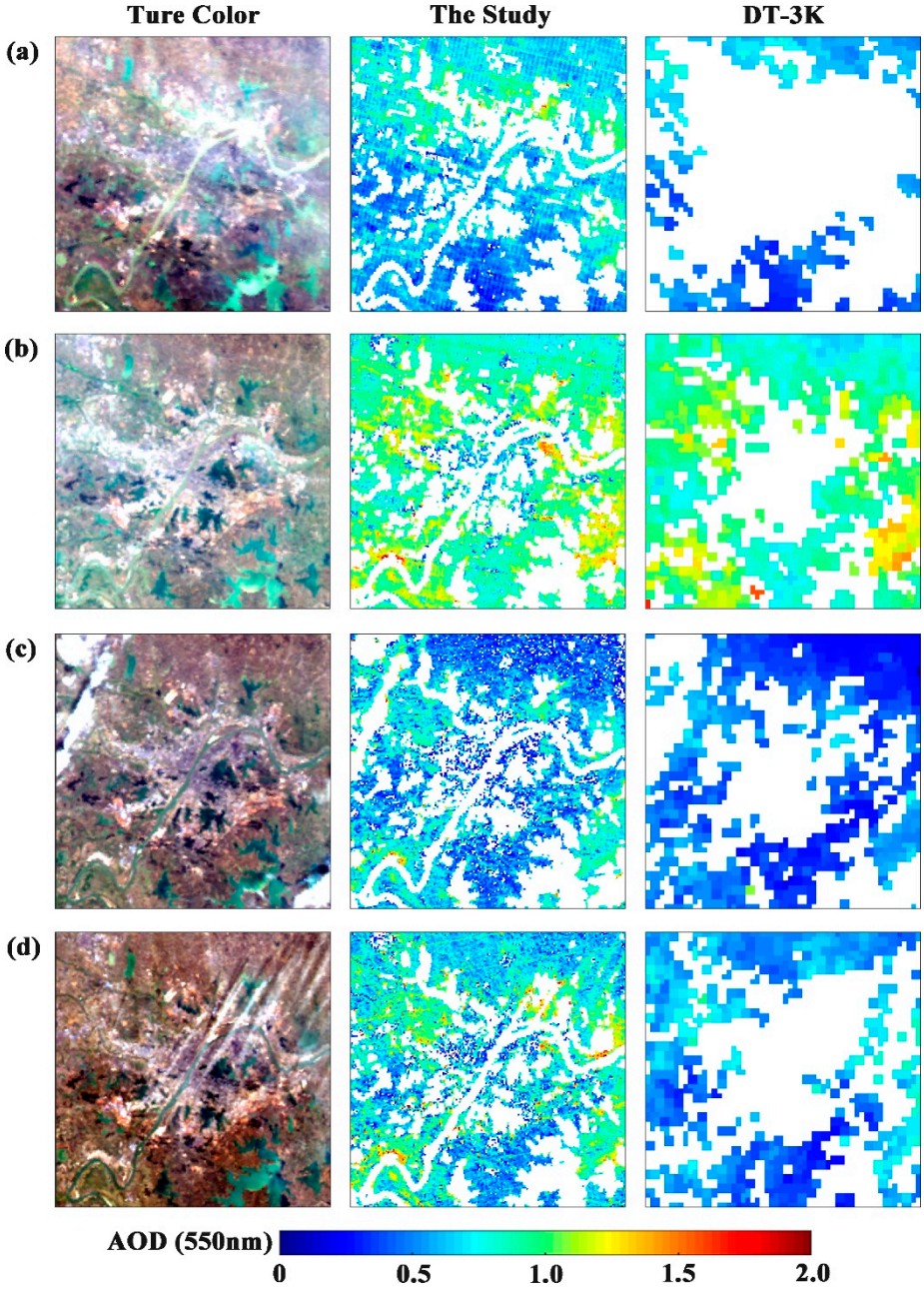

**Figure 6.** Four cases of our 500 m AOD retrieval results versus DT-3K products are selected on Terra and Aqua MODIS sensors: (**a**) 17 December 2018 03:30 UTC, (**b**) 26 February 2017 03:00 UTC, (**c**) 14 February 2017 05:45 UTC, and (**d**) 26 February 2018 05:40 UTC.

## 4. Conclusions

A new method was developed to retrieve AOD at a 500 m spatial resolution from MODIS data collected over urban surfaces under heavy aerosol loading conditions in winter. Local aerosol models were calculated by clustering analysis, based on multiyear measurements performed by a CIMEL sun–sky radiometer. A surface reflectance estimation scheme [37], based on the inverse distance weighting method, was employed and promoted to higher spatial resolution conditions. This scheme was found to perform well in the study area. After pixels screening (for the removal of clouds, snow, water, and shadows), the process of AOD retrieval was executed to match dynamic aerosol models and the values of AOD in the meantime by comparing measured and theoretical apparent reflectance in two MODIS channels. The relatively high resolution AOD retrieval method proposed in this

study supplemented aerosol information over urban surface effectively. Like the well-known MODIS operational DT algorithm, our method can be used in other urban environments with serious air pollution and has the potential to be transplanted in similar other sensors.

The retrieval results were verified with the sun–sky radiometer and the MODIS C6.1 aerosol products. The AOD retrieved in correspondence of the urban surface were well correlated with the data obtained by the sun–sky radiometer. The RMSE and the EE for the Terra satellite were 0.129 and 0.715, respectively; those of the Aqua satellite were 0.181 and 0.704, respectively. R was ~0.89. Moreover, the number of successful retrievals increased significantly, especially under heavy loading aerosol conditions (AOD > 1), when the value of AOD is almost missing in the existing DT algorithm. On a spatial scale, our results are also in good agreement with the MODIS DT products. The mean R values in relation to the data of the Terra and Aqua satellites were 0.905 and 0.847, respectively. In addition, the overestimation of AOD, compared with the MODIS AOD products, was partially alleviated through the use of the improved aerosol models, which considered a higher absorption of the incoming solar light and the rate of particle hygroscopic growth.

The present study is the first of a series of investigations on urban haze and particle pollution supported by the Wuhan government. We improved the accuracy and efficiency of AOD retrieval over an urban surface under the heavy aerosol loading conditions typical of winter. In future works, we will apply this method to FY-3D satellite images with the aim of further investigating air pollution in Central China.

**Author Contributions:** S.J. designed the experiments and prepared the manuscript draft; Y.M., M.Z., and W.G. helped to organize the research, analyze data and revise the manuscript. Y.M. and M.Z. have the same contribution to the study; O.D. advised on the study; B.L., Y.S., and C.Y. assisted to complete the experiments.

**Funding:** This work was supported by the National Key R&D Program of China (Grant No. 2018YFB0504500), the National Natural Science Foundation of China (Grant No. 41875038, No. 41801261, and No. 41905032), the Natural Science Foundation of Hubei Province (Grant No. 2017CFB404), and the Wuhan Science and Technology Plan (Grant No. 2019020701011453).

**Acknowledgments:** We are grateful to the MODIS scientific teams for the provision of MODIS data and products, and the Landsat Science Team for the Landsat-8 OLI image used in this study. Finally, we would also like to thank all anonymous reviewers for their constructive comments.

**Conflicts of Interest:** The authors declare no conflict of interest.

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
