# Peer review of "Retrieval of 500 m Aerosol Optical Depths from MODIS Measurements over Urban Surfaces under Heavy Aerosol Loading Conditions in Winter"

_remotesensing, doi:10.3390/rs11192218_

Round 1

Reviewer 1 Report

The authors have developed an interesting improved aerosol retrieval algorithm applied to MODIS which they have applied to the urban area of Wuhan, China. The application of k-means clustering resulted in two aerosol models, a scattering and an absorbing one (SAM and AAM). Can the authors comment on the dependence of the single scattering albedo on the AOD for both models: What aerosol properties are causing this dependence? The authors have validated their retrievals with a ground-based sun-sky radiometer based in the study area. The description of the method and results is detailed and straight-forward to follow. The paper is scientifically sound. The paper should definitely be published. The paper would benefit from some improvements to its English.

Author Response

Dear reviewer, thanks for your comments. We have revised our manuscript and improved the English expression carefully.

Aerosols are affected by many factors in the study region, such as local meteorological conditions, source of anthropogenic emissions, and long-distance transports of aerosol particle. The dependence of the single scattering albedo on the AOD is mainly due to the differences in composition and microphysical properties of aerosol particles. During periods of air pollution, the fine-mode water-soluble component increased rapidly and reached almost 50% of total sub-micron particles (Cheng et al., 2014). These water-soluble particles showed no-absorbing properties, and their size increased due to hygroscopic growth and thus enhanced the total extinction. However, sometimes aerosol particles showed relatively strong absorbing due to local or transported black carbon (Zhang et al., 2017), which came mainly from winter biomass or fossil fuel burning. Therefore, we used the SAM and the AAM to describe the local aerosols.

1. Cheng, H.R., Gong, W., Wang, Z.W., Zhang, F., Wang, X.M., Lv, X.P., Liu, J., Fu, X.X., Zhang, G. Ionic composition of submicron particles (pm1.0) during the long-lasting haze period in january 2013 in wuhan, central china. J. Environ. Sci., 2014, 26, 810-817.

2. Zhang, M., Ma, Y., Gong, W., Wang, L., Xia, X., Che, H., Hu, B., Liu, B. Aerosol radiative effect in uv, vis, nir, and sw spectra under haze and high-humidity urban conditions. Atmospheric Environment, 2017, 166, 9-21.

Reviewer 2 Report

General comments:

Aerosol is very important to impact atmospheric cycle and climate system by direct and indirect effects, a hot issue of scientific researches internationally. Also, atmospheric pollutions cause adverse harm to human health, such as, aerosols and gases. Aerosols are known to originate from direct emission and secondary formation, namely, POA and SOA. The organic aerosol (OA) is a very important part of aerosols, including BC and OC. Air pollutions always cause an important problem, and become an open issue.

In order to obtain AOD at higher resolution of 500m in urban regions, this paper improves aerosol model compared to MODIS, describes retrieval for AOD method in detail, presents results and validates by observed and retrieved AOD. Overall, the paper has some advance, and obtains some available results. The topic of this paper is of common interest within the scientific community. Although the manuscript includes some important results, however, the quality is not sufficient in the current state to be directly published. The authors should take the suggestions made here into consideration for revision.

Specific comments:

As for aerosol types in Wuhan, the paper should provide more evidence to support SAM and AAM, by comparing with local measurements on the ground. As for validation of AOD retrievals, the paper should give more estimations, such as comparison of retrieved 500m AODs and sun-sky measurements, validations with measurements at different time scales, e.g. monthly, seasonal, annual.

Author Response

Dear reviewer, thanks for your comments. The SAM and the AAM are calculated by K-Means clustering method using measurements of a CIMEL sun–sky radiometer (CE-318). The CE-318 is generally recognized as the most accurate ground-based aerosol observation instrument. Our CE-318 has been calibrated annually using the China Meteorological Administration Aerosol Remote Sensing Network (CARSNET) which is an official calibration platform (Che et al., 2009). The calibration can be mainly divided into two steps: 1) indoor correction by the standard laboratory integrating sphere; 2) joint observation with standard CIMEL sun photometers which are installed in China Meteorological Administration. These standard CIMEL sun photometers were calibrated using the Photome trie pour le Traitement Ope rationnel de Normalization Satellitaire (PHOTONS) calibration facilities in Lille (France), and Carpentras (METEOFRANCE). And the PHOTONS master instrument was calibrated by Langley plot analysis at Izana Observatory (AEMET, Spain) following the calibration protocol used by NASA staff. Therefore, our measurements are reliable.

Retrieval results were verified in this study by comparing with in-situ measurements and MODIS AOD products (C6.1). Terra and Aqua satellites observed the region only once or twice a day and had only few data available caused by high cloud cover and frequent haze events. Therefore, the validations at different time scales is difficult due to limited data.

In addition, we have also improved the English expression in the manuscript carefully, for readers reading.

Che, H.Z., Zhang, X.Y., Chen, H.B., Damiri, B., Goloub, P., Li, Z.Q., Zhang, X.C., Wei, Y., Zhou, H.G., Dong, F., et al. Instrument calibration and aerosol optical depth validation of the china aerosol remote sensing network. J. Geophys. Res.-Atmos., 2009, 114, 12.